# Determining the Enablers and Barriers for the Adoption of Clean Cookstoves in the Middle Belt of Ghana—A Qualitative Study

**DOI:** 10.3390/ijerph16071207

**Published:** 2019-04-04

**Authors:** Francis Agbokey, Rebecca Dwommoh, Theresa Tawiah, Kenneth Ayuurebobi Ae-Ngibise, Mohammed Nuhu Mujtaba, Daniel Carrion, Martha Ali Abdulai, Samuel Afari-Asiedu, Seth Owusu-Agyei, Kwaku Poku Asante, Darby W. Jack

**Affiliations:** 1Kintampo Health Research Centre, Post Office Box 200, Kintampo, Brong Ahafo Region, Ghana; francis.agbokey@kintampo-hrc.org (F.A.); rebecca.dwommoh@kintampo-hrc.org (R.D.); theresa.tawiah@kintampo-hrc.org (T.T.); kenneth.ae-ngibise@kintampo-hrc.org (K.A.A.-N.); mohammed.mujtaba@kintampo-hrc.org (M.N.M.); martha.abdulai@kintampo-hrc.org (M.A.A.); samuel.afari-asiedu@kintampo-hrc.org (S.A.-A.); sowusuagyei@uhas.edu.gh (S.O.-A.); 2Mailman School of Public Health, Columbia University, Department of Environmental Health Sciences, New York, NY 10032, USA; dc2846@cumc.columbia.edu (D.C.); dj2183@columbia.edu (D.W.J.); 3Institute of Health Research, University of Health and Allied Sciences, PMB 31, Ho, Volta Region, Ghana

**Keywords:** clean cookstove, adoption, enablers, barriers, middle belt, Ghana

## Abstract

Despite its benefits and espousal in developed counties, the adoption of clean cookstoves is reportedly low in less developed countries, especially in Sub–Saharan Africa. This qualitative study aimed at exploring and documenting the enablers and barriers for adoption of clean cookstove in the middle belt of Ghana. The findings showed convenience of clean cookstove use, reduced firewood usage, less smoke emission and associated health problems resulting from indoor air pollution and time for firewood gathering and cooking, good smell and taste of food as enabling factors for clean cookstove adoption. Factors such as safety, financial constraint (cost), non-availability of spare parts on the open market to replace faulty stove accessories, stove size and household size were the potential barriers to clean cookstove adoption. These findings help us to understand the factors promoting and inhibiting the adoption of clean cook stoves, especially in rural settings.

## 1. Introduction

Household air pollution (HAP) is a risk factor for a number of health-related conditions including pneumonia in children, cardiovascular disease, and chronic obstructive lung disease for both males and females globally [1,2]. HAP was ranked the fourth leading health risk factor and was linked to close to 4.3 million deaths in 2012 and to 2.6 million deaths in 2016 [3,4]. Cooking with solid fuels such as dung, charcoal, wood, plants and crop residue over open fires or in simple traditional cookstoves expose household members to high pollutant concentrations [5]. The use of biomass fuels has been shown to result in adverse social impacts such as time spent collecting firewood [6], and to and environmental impacts such as ambient air pollution [7,8], and deforestation [9]. In spite of this, almost a third of the world’s population—three (3) billion people worldwide continue to rely on solid fuels and traditional cookstoves for their heating and cooking needs [5,10]. Over 76% of sub-Saharan Africa’s population rely on traditional biomass as their main energy source [11].

The Sustainable Development Goals (SDGs) advocate access to affordable, reliable, sustainable and modern energy for all [12]. In response to this and other calls, several efforts to promote the adoption of clean cookstoves are underway to alleviate the adverse effects of HAP. For example, The Clean Cooking Alliance was created to promote household adoption of clean cookstove and fuels globally [13]. Locally, government has responded to the call by developing policies and programs to address the problem of HAP. In Ghana, the government has distributed 150,000 liquefied petroleum gas (LPG) cookstoves to residents in rural communities via the Rural LPG Programme.

The adoption of clean cookstoves for cooking by households has been found to be likely to reduce biomass fuel usage, reduce exposure to HAP and potentially save households time from firewood gathering and cooking [14,15]. However, initiatives that promote adoption of clean cookstove use often face hitches such as households’ insufficient or unsustainable fuel use, and poor stove maintenance [16,17].

Kintampo Health Research Centre, in collaboration with the Columbia University conducted the Ghana Randomised Air Pollution and Health Study (GRAPHS) in the Brong-Ahafo Region of Ghana to assess the health impact of using clean cookstoves on birth weight and pneumonia [18]. In this study, clean fuels, LPG and clean BioLite cookstoves (BioLite Inc. Brooklyn, NY, USA) shown in Figure 1 below, were supplied to participants in the intervention study. The LPG stoves provided had very low emissions [19]. Study participants were followed up from pregnancy to one year after delivery. Those in the LPG group were supplied with LPG throughout the duration of GRAPHS but gas delivery was discontinued after they exited the study. Participants in both the LPG and BioLite groups were however expected to continue the use of their stoves after they exited the study. Overall stove use during the GRAPHS was high. Out of the 360 participants assigned to the LPG arm, one participant (0.3%) did not use the LPG stove at all over the 52-week period of follow-up after delivery; whilst 277/360 (77.0%) used it more than 80% of the time during the follow-up. Within the BioLite arm, less than 1% (4/521) participants never used their BioLite cookstoves over the 52-week period of follow-up after delivery; whilst 40.5% (211/521) used it more than 80% of the time during the follow-up period.

A qualitative study was conducted to determine the enablers and barriers for the adoption of clean cookstoves among participants of the (GRAPHS) for a period of one year after they had exited the study. 

## 2. Materials and Methods

### 2.1. Study Settings

The study was conducted in the Kintampo north and south districts in the Brong-Ahafo Region of Ghana. With a resident population of 154,341 at the last count (Kintampo Health Research Centre, 2017), the area is predominantly rural and is located in the forest-savannah transition zone in the middle belt of Ghana. The primary cookstove of households in the study area is the traditional 3-stone firewood stove [18] as shown in Figure 1 above. 

The main fuel used by households in the area is wood, although charcoal is also used. There are two seasons in the area, wet and dry. Cooking is generally done outdoors during the dry season but during the wet season, cooking is done in enclosed or covered kitchens. [20]. GRAPHS had long-term users of clean cookstove in the two districts who were followed up for a year with primary focus on its impact on birthweight and respiratory health of children [18]. The eligibility criteria for participation by a woman in the original GRAPHS were that she is the primary cook in her household, is pregnant and in a gestational age of ≤24 weeks, and the pregnancy should be a singleton fetus. Randomization was carried out at the village level. All intervention villages were given clean cookstoves—either BioLite or LPG.

### 2.2. Study Design

This study is hinged on the methodological orientation of the grounded theory. In using this theory, qualitative data was collected and analyzed based on themes [21].

### 2.3. Participants’ Selection

One hundred and thirteen (113) women who had exited at least one year from the intervention (BioLite and LPG) arms of GRAPHS prior to commencement of this study were purposively selected for the focus group discussions (FGD). Participants included 59 LPG and 54 BioLite users. 

### 2.4. Data Collection

Interview guides were developed and piloted by the research team. Ten (10) FGDs (five each for BioLite and LPG arms) were conducted between August and September 2015. All FGDs were successfully completed at the first attempt and no repeated interviews were done. The purpose of the study was explained to study participants, after which the written study information sheet was read and explained by members of the research team. Additionally, some background socio-demographic information was recorded. A copy of the signed consent form was given to each participant before the actual FGD took place. There were eight to twelve (8–12) participants with similar characteristics in each discussion group. The FGDs were conducted in the Twi language, which is the primary spoken local language in the study area. All sessions were tape recorded. Field notes were taken by a note taker to complement the audio recordings, in order to capture body gestures and facial cues expressed by participants during the discussions. Each FGD session lasted approximately ninety (90) min. Data collection was brought to a close at the point of saturation where no new information was being added upon reaching the ninth and tenth FGD sessions.

### 2.5. Data Management and Analysis 

Each audio recording was transcribed verbatim and translated from the local Twi language and typed with appropriate quality control checks into Microsoft word 2016 (Microsoft Corporation, Redmond, WA, United States). QSR NVivo qualitative analysis software Version 10.0 (QSR International, Melbourne, Australia) was used in highlighting common themes both a priori and posteriori. The results were presented in themes and supported by quotes. All FGDs were done in a serene environment either under trees, in a church, health centre or school in the participants’ home community. Aside from the team of researchers and participants selected for the sessions, there were no other persons present during the FGDs. This was to ensure that participants did not feel intimidated by the presence of bystanders. 

### 2.6. Ethics

The study was approved by the Kintampo Health Research Centre Ethics Review Committee with reference number (KHRCIEC/2015-6). Written consent was obtained from all FGD participants before their participation in the discussions. 

## 3. Results

Participants were predominantly engaged in peasant farming 65% (73/113). The average number of children per participant was 4, whilst the average household size was five (5) as shown in Table 1 above. We note that the LPG arm participants had a higher percentage of participants who have completed basic education. We believe that this was an artifact of the small sample size of this qualitative study. Since our goal in this study is not to compare across study arms, we are not concerned that this difference introduced bias. 

Prior to receiving the intervention clean cookstoves from GRAPHS, participants in both the BioLite and LPG arms generally reported relying primarily on traditional hearth (three-stone) which used fuels such things as firewood, dung and other biomass residues for cooking. This is illustrated by the following response from a participant in the BioLite arm:
*I was using the traditional three-stone stove* (FGD 1; Participant #7). Some others also reported using coal pot which uses charcoal, as indicated in Table 1 above.


### 3.1. Enabling Factors for the Adoption of Clean Cookstove

A number of the participants reported a number of factors they considered as the enablers or motivators to the adoption of clean cookstoves; and based on themes identified as most important after coding using the qualitative analysis software (NVivo version 10.0, (QSR International, Melbourne, Australia). These include the following:

**It cooks faster:** responding to why they continue to use their clean cookstove, most of the participants said they continued to use it because it cooks faster compared to the traditional 3-stone thus, saving time for other activities. Participants in the LPG arm had this to say:
‘*It [LPG] helps my children. When they wake up in the morning, I am able to do things quickly for them to go to school early. It is helpful to the mothers in terms of cooking. If you want to go somewhere it can help you to cook quickly so that you can do your work*’.(FGD 3; Participant #3)
‘*As you can see, I have gained weight, and looking good and healthy. It is the gas that made me gain weight. Whatever you want to cook, even ampesi [local dish], right now if I go home and cook anything, it won’t take long to cook and I will get time to do what I want. I can use that time to clean my room, even if it is bathing, I can do that and by the time I come back, the food would have been cooked. If it is yam [local dish], I won’t even add pepper, I will just eat it like that and drink water and I will be full. If it is firewood, I would have to go around to look for fire. If I don’t get some and I don’t have matches, I just have to stop the cooking and go and buy food from outside, even some kenkey [local dish] and just come and eat it like that.*’(FGD 4; Participant #4)


A participant in the BioLite arm had this to say about the BioLite:
*I cooked fufu with it today and the fufu took just a little while to cook*.[FGD 5; Participant # 8]


**Reduced quantity of firewood consumption and time spent on gathering firewood**: the traditional cookstove used by women in most households in the study area is an open fire with three large stones placed together to hold cooking pots. Most of the participants in this study reported that the traditional cookstove required a lot of firewood and that gathering enough firewood was very time consuming and, in some cases expensive and quite difficult. This they said resulted in household members particularly women and children having to spend a lot of time and energy in gathering firewood to meet the heating and cooking needs of the household. With the introduction of the intervention cookstoves, less or no time and money is spent in gathering or on purchasing firewood. Whilst the LPG stove uses no firewood, the BioLite cookstove requires lesser quantity of firewood which is chopped into smaller pieces for cooking compared to the traditional 3-stone cookstoves. Participants in the BioLite arm had this to say:
‘*BioLite cook stove uses less firewood. You can use the bundle for up to four days. But with the three-stone-stove, the most number of days that the same bundle of firewood can last is maybe two days*’.(FGD 6; Participant #5)

A participant in the LPG arm also had this to say:
“*It is really helpful. I really get time. I can put food on fire and if I want to wash too, I can do that as well. If it was the muchia, [traditional 3-stone stove] I won’t get the time to do all that. If you put the food on, you have to be pushing the firewood into the fire and fan it. There is also smoke. With all this, can you do anything else whilst cooking? No, you can’t [referring to the traditional stoves]*”.(FGD 9; Participant #10 LPG arm)


**Allows for household multitasking**: the use of the clean cook stoves allows for multitasking. Participants generally reported that they are able to attend to other household chores such as washing of dishes, sweeping or bathing whilst cooking at the same time which hitherto, couldn’t have been done with the 3-stone. This is indicated by the quote below from an LPG participant:
“*What motivates me to use it is that when it is late and I am cooking, I put the meal on one stove and the soup on the other so I don’t get tired. Even days that I cook soup, I can put the soup on one stove and the meal on the other stove, and bathe the children whilst the food is on fire cooking*.”(FGD 7; Participant #7)

A participant from the BioLite arm had this to say about the BioLite stove:
“*Now I have more time to do other things because the coal pot cooks [BioLite] food faster than the three-stone-stove. And I use much less fire*.”FGD


**Taste of food:** participants generally were of the view that food prepared on the CC tastes better and devoid of smell of smoke compared to the 3-stone. This is illustrated by the following quotes:
‘*The gas [LPG stove] has been very useful to me. It helps me to be fast with my cooking. My food also smells good. The food does not smell of smoke compared to the ‘muchia’ [traditional 3-stone stove]*’.(FGD 4; Participant #8)
‘*Whenever I prepare soup on the traditional stove, the smoke enters the soup but the BioLite cook stove is smokeless and therefore does not give the soup a bad taste*’.(FGD 5; Participant #7)
*The smell of the 3-stone stove is very much like smoke, and when you’re eating you taste the smoke. With the BioLite cook stove that is not the case*.(FGD 8; Participant #6)


**Less smoke = good health:** participants recounted that the use of the intervention cookstove promotes good health as it emits a little smoke in the case of the BioLite stove or no smoke at all in the case of the LPG, compared to the 3-stone. According to participants, this reduces their chance of contracting smoke-related diseases:
‘*It is very helpful to us. Since the LPG was given to us, we cook well without inhaling any smoke. Our babies are also very healthy*.’(FGD 3; Participant #1, LPG arm)
‘*What motivates us is that as we use it, it protects us and our children from smoke-related illnesses and gives us good health. It protects us in the sense that the extent to which we were inhaling smoke from the ‘muchia’, now we don’t inhale it any more*’.(FGD 7; Participant #8)


**Promotes harmony in the home:** most participants reported that the use of the LPG cookstove has brought about peace and harmony in their homes. Husbands are able to help cook for the household when the woman is not in the position to cook due to ill health or absence. This is because the stove can be used indoors unlike the 3-stone which is used outside thereby making the men shy to cook outside lest they are seen and ridiculed. Participants in the LPG arm had this to say:
“*When you are sick, sometimes you are not able to sit down to cook. Your husband is able to do the cooking on the gas. Previously, he couldn’t use the ‘muchia’ [traditional 3-stone stove] because of the smoke. He was even shy to cook outside; I was shy to allow him to do that. If the stove is in the veranda and he uses it, no one will see him doing it and it doesn’t take long for him to finish*.”(FGD 3; Participant #6, LPG arm)
“*The gas is really useful. Previously when we were using the muchia, I could work and in the evening, I would be too tired to do anything but would still have to wake up early to go to the farm for firewood. Early in the morning by the time my husband will ‘need’ me, I would have gone to the farm for firewood. By the time I come back my husband will be angry because I woke up too early to go to the farm for firewood. With the gas, I wake up early and put water on fire, by the time I finish ‘receiving it’ from him [referring to sexual intercourse] the water would be ready for him to bath*.” (FGD 4; Participant #5, LPG arm)


This view is shared by participants who use the BioLite stove:
“*Because of its ability to cook two different meals at a go, I am able to finish through with my cooking on time and have time for other things such as chat with my children and husband*.” (FGD 1; Participant #11, BioLite arm)


### 3.2. Perceived Barriers to the Adoption of Cleaner Cookstoves

***Economic Barriers:*** financial constraints generally emerged as one of the key barriers to the sustainable use of LPG by most of the respondents. Respondents indicated that they do not have jobs and regular sources of income and are not be able to refill the LPG cylinders for continued use after the GRAPHS trial ends and free LPG deliveries cease. The responses below support this finding.
“*The benefits are good but the financial problem will be the thing. We farm only yearly. There are no jobs here so we are unemployed. This year like this, the rains didn’t come so if the groundnuts go bad that will be it until next year*.”(FGD 3; Participant #8)
“*It is money issues that will prevent us from refilling after the study. It cooks well and smoke doesn’t enter the food*.”(FGD 5; Participant #10)


In this regard, some respondents further specified categorically that they have no alternative but to return to the use of the traditional 3-stone fires. This is what a respondent had to say in support of this finding:
“*I wish I will continue to use it but I won’t get the money… I will go back to use the muchia (traditional 3-stone fires) and my disease will start again. My husband will not help me with the refill. Once he knows that there is firewood, he won’t give me money to refill the gas*.”(FGD, 7; Participant #6)


Some respondents mentioned that they may be able to refill their cylinders after the study but not on regular basis. Respondents were ready to refill their cylinders for use when money is available. As some respondents explained, they have other responsibilities such as wards’ school fees to attend to and until they are able to pay for all these, they cannot refill their cylinders. This finding is typically noted in the response below:
“*We have school children. When they ask for fees, we can’t use the money to refill the gas at the expense of their fees. I will go back to reuse the muchia [traditional 3-stone fires] until I get money to refill the cylinders*.”(FGD 3; participant #5, LPG arm)


Due to financial constraints some respondents have also decided to sell one of their cylinders since they will not be able to keep the two. The proceeds from the sale of the one cylinder will be used to purchase LPG. After this, the cylinder will be refilled as and when respondents obtain money. To this end, this is what a respondent had to say at an FGD session.
“*I had made up my mind to sell one of the cylinders and keep one after the study. I won’t get money to refill both of them.so I will sell one and keep one. I hear the cylinder is about GH¢50.00 or GH¢ 60.00; I can refill the cylinder once or twice. When I get money then I go and refill it again*.”(FGD 3; Participant #8, LPG arm)


A few participants notwithstanding the concerns of others about their inability to overcome the financial constraint of refilling their cylinders, expressed their willingness of continuing with the refill. A participant had this to say:
“*No, cost will not be a problem. I can refill it. Whatever amount that I get and will go and refill it and be using small, small without allowing any other persons to use it with me*” (FGD 4; Participant #3, LPG arm)


Whilst financial constraint was the main barrier in getting fuel for the LPG stove, it was not same for those households who use BioLite stoves since they usually gathered the firewood free of charge and did not have to buy fuel for cooking. Participants in the BioLite arm were more concerned about getting the required size of firewood/sticks needed for the stove. Unlike the 3-stone stoves that can use any firewood irrespective of the size, the BioLite stove requires smaller pieces of firewood/sticks to lite. Participants therefore had to take time to chop the firewood into smaller pieces before it can be used in the BioLite stove. A participant in the BioLite arm had this say:
“*The BioLite cook stove consumes less firewood as compared to the traditional stove. I can split one firewood into bits and cook a full meal with it but with the traditional stove, I have to spread a lot of firewood on each of the three sides of the stoves for food to cook thoroughly. However, with the traditional stove, I have to go fetch a new bundle of firewood after every three days of fetching them. However, with the BioLite cook stove, I can use the firewood for a week before I go fetch a new bundle*.” (FGD 7; Participant #3, BioLite arm)


**Safety concerns:** Some people in the communities perceive LPG to be the cause of burns and fires in houses and so are not enthused about using them in their homes. Whilst some women were encouraged to be part of the study by their husbands and households, some respondents alluded that other women may have been discouraged by their husbands and household heads due to their perception of the safety of the LPG stove stoves. A respondent had this to say:
“*Some men may not be happy with it. They can tell the women not to bring in the gas because it can burn down the house and the children*.”(FGD 4; Participant #6, LPG arm).
“*My child had been burned before. It happened in my absence. I had food on fire and his siblings looked on unconcerned whilst he moved close to the stove and touched it and got burned*.” (FGD 8; Participant #1)


**Size of the stove:** Though participants reported the continuous use of their BioLite cook stove, they indicated that they still use their traditional hearth cook stove alongside. Some related this behavior to the size of the stoves which they considered too small to support big pots, which they use when cooking for a larger number of people. The women in the study communities usually cooked for larger number of household members as well as farm laborers. This concern was mostly shared by those in the BioLite arm. A participant had this to say:
*There will be no problem if we are four and I’m using the BioLite cook stove because a smaller pot size will be required to do that and which can fit the size of the stove, but it’ll be hard for me to keep using it if we become 10 and I have to cook to feed all. The large pot can’t fit properly on it, especially for abetie so the best option is to shift to the 3-stone for that*. (FGD 8; Participant #8, BioLite arm)


A participant in the LPG arm had this to say:
*I cook kenkey (a local meal) so I go for firewood to set fire with the 3-stone stove. The cooking pot is very big and cannot fit on the LPG stove and so cannot cook the meal. I do that on a weekly basis*. (FGD 4; Participant #4, LPG arm)


**Maintenance and repairs:** Furthermore, participants cited frequent breakdown and malfunction of the BioLite cook stove and delayed repaired works as something that could serve as a barrier to getting a lot more women to use the stove. A participant had this to say:
*When the stove develops a fault, I am unable to use it to prepare food. I have to send it for repairs in order to start cooking with it again. This has happened on two occasions*. (FGD 10; Participant #1, BioLite arm)
*The only barrier will be when the stove ceases to work. Mine has broken down on five occasions*.(FGD 10; Participant #2, BioLite Arm)
“*If the stove gets spoilt and it isn’t repaired, you’ll be motivated to go back and use the 3-stone stove*.”(FGD 8; Participant #1, BioLite Arm)


### 3.3. Misconceptions about the Study

There was an initial misconception that blood will be drawn from people who agree to be part of the study. Some people were, therefore, advised by friends, neighbors or relatives not to accept the LPG because it is an attempt by the KHRC to lure them to take their blood.
“*When the program started, some people said that (KHRC) were going to take blood from our babies, but ever since it started, no one has taken my child’s blood*.”(FGD7, Participant #3, BioLite arm)


Notwithstanding this, some respondents mentioned that they were encouraged by other people and relatives such as their husbands to be part of the study. The excerpt below illustrates this finding:
“*When the gas came I wanted to take one but my friends told me they will be taking blood so I wasn’t going to take it. But my husband told me to take it because I will be taken care of if I take it and that he won’t get money to take care of me if I don’t take it. So, I went for it*.”(FGD 3; Participant #5, LPG arm)


## 4. Discussion

Knowledge and understanding of the enabling factors and barriers to the adoption of clean cookstoves is important for rolling out new technologies in the clean cookstove industry that will be adopted by the people for whom it is intended. 

Findings from this study indicate that household size and cookstove size were factors that are key to influencing the adoption or otherwise of clean cookstove. Participants with large household sizes reported having to combine the use of the clean cookstove (either LPG or BioLite) with the traditional cookstove. This is because preparing certain types of meals like *“banku” or “abitie” (which requires that more strength is exerted to prepare it)* for a large household calls for the use of a big cooking pot, the sizes of which generally tend to be too big for the size of the clean cookstoves to support. This finding agrees with a similar one reported by a study undertaken in rural Peru where household size and stove size were reported as key factors for determining adoption of clean cookstoves [13]. 

Availability of people within the community with the ability to repair broken clean cookstoves resulting from regular use, and how readily available and affordable spare parts are to households within the community, were found to be factors that will either motivate or discourage the adoption of clean cookstoves. For instance, some participants who had their stoves or an accessory like regulator (for LPG users) malfunction will have to return to use the traditional 3-stone stove until they were fixed or replaced by the study team. However, in case a similar thing was to happen after the study they would find it difficult replacing it due to financial constraints. In the case of clean BioLite cookstove users, absence of a means of repair of broken stoves resulted in households reverting to using their traditional 3-stone stove, thus making the adoption of cleaner cooking systems incomplete. These findings are in line with another study conducted in peri-urban Kiambu County, Kenya and urban Lusaka, Zambia which reported similar findings [22]. 

Also, as reported in this study, the time saved in cooking and gathering firewood for cooking which impacts on women’s time, goes a long way in serving as a motivation for the adoption of clean cookstoves, from the user of the LPG cookstove who does not spend any time on firewood gathering to the user of the BioLite cookstove who spends less time vis-à-vis what would have been spent if it was meant for the traditional 3-stone stove, in gathering firewood [23,24,25]. The time saved on cooking or firewood gathering enables women to undertake other household chores whilst cooking or investing in other income-generating activities [26] 

Furthermore, the findings suggest that participants were prepared to continue using the clean cookstoves in place of their traditional 3-stone stove provided they would be receiving free LPG due to financial constraints. This presupposes that people are more likely to adopt clean cookstoves as long as it comes to them at no financial cost as reported by other studies [27]. Also, how readily available and affordable LPG is will go a long way in determining if clean cookstoves (using LPG) will be adopted, and the extent to which this adoption is done as found and reported by a rural Peruvian study [13].

Moreover, safety has the potential of determining the uptake of clean cookstoves. When people perceive that using clean cookstoves expose them and their household members and property to danger, they are more likely not to use them [26].

### Limitation

Although women are generally the primary household cooks in the study area, interviewing men to get their thoughts on the subject matter would have given a more holistic perspective on the subject. Furthermore, cookstoves and LPG fuel were provided for free, thus limiting external validity. Also influencing external validity is the fact that this study was limited to one region of Ghana. Moreover, there was no attempt made by this study to match self-reported stove use to stove use monitors (SUMs) data. Again, the generalizability and results of this study may be particularly limited for issues related to cost. 

## 5. Conclusions

Adoption of clean cookstove in the middle belt of Ghana seems to resonate well with households following the free supply of clean cookstoves by the GRAPHS. However, with regard to clean cookstoves, despite their advantages reported by this study, beneficiary households have not totally replaced the traditional 3-stone stove which was still being used alongside the clean intervention cookstoves. The primary enablers include: ability to cook faster, taste of food, reduced quantity of firewood consumption and time spent on gathering firewood, ability to multi-task, and emission of less smoke equalling good health. The barriers include: cost of fuel, stove size, as well as maintenance and repairs. Findings from this study adds to the growing body of evidence on factors encouraging or discouraging adoption of clean cookstoves and further helps in guiding the design of clean cookstove interventions going forward. 

## Figures and Tables

**Figure 1 ijerph-16-01207-f001:**
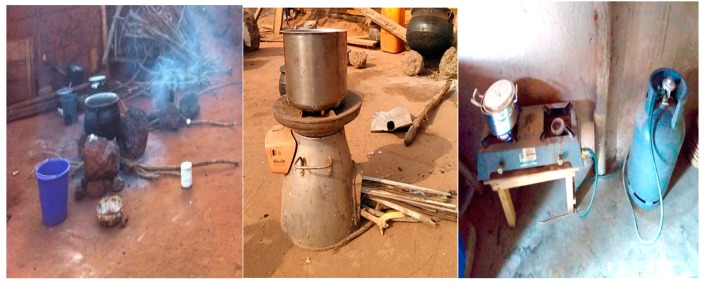
Three-stone (**left**), BioLite (**middle**) and (**right**) liquefied petroleum gas (LPG) cookstoves.

**Table 1 ijerph-16-01207-t001:** Socio-demographic characteristics of respondents.

Demographic Characteristics	Summary Statistics (*N* = 113)	Summary Statistics LPG Arm (*N* = 59)	Summary Statistics BioLite Arm (*N* = 54)
Age; *mean* (SD)	40 (5.6)	40 (5.6)	40 (5.6)
Occupation; *n* (%)			
Unemployed	7 (6.2%)	4 (6.8%)	3 (5.5%)
Farming	68 (60.2%)	35 (59.3%)	33 (61.1%)
‘Petty’ Trading	28 (24.8%)	13 (22%)	15 (27.8%)
Other	10 (8.8%)	7 (11.9%)	3 (5.5%)
Educational Level; *n (%)*			
No formal education	86 (76.1%)	40 (67.8%)	46 (85.2%)
Completed at least Basic Education	27 (23.9%)	19 (32.2%)	8 (14.8%)
Marital Status; *n (%)*			
Single	29 (25.7%)	15 (25.4%)	14 (26.0%)
Married	84 (74.3%)	44 (74.6%)	40 (74.0%)
Religion; *n (%)*			
Christianity	86 (76.1%)	42 (71.2%)	44 (81.5%)
Islam	25 (22.1%)	15 (25.4%)	10 (18.5%)
Other	2 (1.8%)	2 (3.4%)	0 (0.0%)
Ethnicity; *n (%)*			
Akan/Bono	56 (49.6%)	30 (50.8%)	26 (48.1%)
Mo	21 (18.6%)	10 (17.0%)	11 (20.4%)
Bimoba/Konkomba/Dagomba/Gonja	27 (23.9%)	13 (22.0%)	14 (26.0%)
Other	9 (7.9%)	6 (10.2%)	3 (5.5%)
Specific fuels used at baseline; *n (%)*			
Firewood	99 (87.6%)	51 (86.4%)	48 (88.9%)
charcoal	11 (9.7%)	7 (11.9%)	4 (7.4%)
Other	3 (2.7%)	1 (1.7%)	2 (3.7%)
Household Size ranged from:	1–19 with an average of 5	1–10 with an average of 3	4–19 with an average of 7

SD = standard deviation.

## Data Availability

Data used for this paper is available upon request through the corresponding author.

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
