# Peer review of "Determining the Enablers and Barriers for the Adoption of Clean Cookstoves in the Middle Belt of Ghana—A Qualitative Study"

_ijerph, 2019, doi:10.3390/ijerph16071207_

Round 1
Reviewer 1 Report
This manuscript focuses on an important topic globally and leverages a strong randomized trial. The manuscript is generally clearly written. I have some questions about the methods and objectives (see detailed comments below). Further, I think the This manuscript focuses on an important topic globally and leverages a strong randomized trial. The manuscript is generally clearly written. I have some questions about the methods and objectives (see detailed comments below). Further, I think the manuscript would be improved considerably if the authors could clearly articulate how the information gained from this study might be used. And finally, the message might be more clear here if the enablers / barriers were connected in some way to measured use / air pollution measurements so that we can see if they do in fact matter in terms of use and exposure.
1. Abstract, line 19: Is there evidence (references) that this is especially a problem in sub-Saharan Africa? Not discussed in
2. Introduction should be updated with most recent GBD numbers. Similarly – the Global Alliance for Clean Cookstoves has changed it’s name (Clean Cooking Alliance).
3. The Introduction is choppy and difficult to follow. Further, the objective of the manuscript/study are not stated in the Introduction.
4. More information is needed in the Methods about the original study in order to understand who is in the current study. What were the eligibility criteria? How did randomization occur? Was randomization successful? How many participants in each study arm were lost to follow-up (not eligible for the current study) and why?
5. How did selection take place for the current study? How many from each arm of the study (BioLite and LPG)? Eligibility criteria? What does “based on geography to ensure a representative sample” mean?
6. Figure 1: Is this proper use of the BioLite stove (large pieces of wood)? Is the LPG stove on the ground? This is generally not considered safe for LPG (stove should be higher than the cylinder)?
7. Line 96: define FGD.
8. Were there further ethics approvals from all institutions involved?
9. The flow of the results would be more clear if the results for BioLite were separated from the results for LPG.
10. Would be helpful to see Table 1 for BioLite and LPG separately.
11. Also – more information about fuel at baseline would be helpful – perhaps in Table 1 (specific fuels).
12. What was the overall stove use in the GRAPHS study?
13. Did you evaluate the connection between the perceived barriers and enablers with actual stove use? Or exposure to air pollution? If not – what how do we evaluate how important the barriers actually were for use?
14. Limitations – should emphasize that the generalizability and results may have been particularly limited for issues related to cost.
15. The Conclusions section is not very clear. Line 363 – what does “caught on well” mean? And line 365-366 – this information has not been discussed previously in manuscript (“have not totally replaced…”). And for this section – how would this information on barriers and enablers be used – particularly given the limitations resulting from the free supply of fuel and stoves in this study?
Author Response
Please find attached point by point response to reviewer 1 comments. Thank you

Reviewer 2 Report
This study investigated the enablers and barriers for the adoption of clean cookstoves in the middle belt of Ghana. The topic is important and interesting. The results can be useful in policy making of promoting clean cookstoves in Ghana.
A major issue of this paper is the presentation of the results. In the present paper, the responses from the interviewees were directly quoted. This is not a scientific way of presenting a systematical survey. Some sorts of statistical method should be used to present the scientific results and draw the corresponding conclusions.
Author Response
please find attached point by point responses to reviewer 2 comments. Thank you

Round 2
Reviewer 1 Report
The revised manuscript is improved in terms of clarity; I thank the authors for their responsiveness. However, I still have concerns about the design, particularly in the selection of the participants. Further, I think the reported enablers and barriers are fairly limited in use if this information is not connected to actual use.
In the Introduction – it does not make sense to include the GBD numbers from both 2012 and 2016 without an explanation. The ranking is different as well. Further, some of the references do not make sense; for example, in the first sentence of the Intro – the reference here for CVD is for the India GBD only. And why are only these health outcomes listed?
Lines 35-37: does not mention wood and charcoal
Lines 39-41: this number (3 billion) is for solid fuels – not just for biomass.
Intro – define LPG on first use.
Line 59-60: make it clear that the emissions were lab-based emissions in a separate study; not clear to me what the next statement about Biolite emissions was based on.
Line 91-92: How is this information about decile of stove use relevant here? How was stove use measured (for this sentence but also for the information reported in Conclusions?
Lines 97-104 are still not clear. What does "purposively selected based on their homogeneous characteristics and ethnicity" mean? How were the 5 groups per arm created / selected?
It would be useful to know how similar the participants/households selected for this study are to those in the GRAPHS study.
In Table 1 – there are some differences between the LPG and BioLite groups. This should be discussed in terms of impact on the results reported here.
The Conclusions section still seems strong given the limitations discussed (and those that I have commented on) – particularly connecting the enablers / barriers to actual use.
Author Response
Comment number | Review comments
| Response |
1. | Lines 35-37: does not mention wood and charcoal
| charcoal, wood have mentioned now as suggested and can be found on line 36. |
2. | Lines 39-41: this number (3 billion) is for solid fuels – not just for biomass.
| This has been changed from biomass to solid fuels as suggested and highlighted green on line 41. |
3. | Intro – define LPG on first use.
| LPG has been defined as liquefied petroleum gas on lines 48 – 49 and highlighted green. |
4. | Line 59-60: make it clear that the emissions were lab-based emissions in a separate study; not clear to me what the next statement about Biolite emissions was based on.
| The statement about Biolite emissions has been deleted from the main text. |
5. | Line 91-92: How is this information about decile of stove use relevant here? How was stove use measured (for this sentence but also for the information reported in Conclusions?
| The statement about decile has been expunged from the text |
6. | Lines 97-104 are still not clear. What does "purposively selected based on their homogeneous characteristics and ethnicity" mean? How were the 5 groups per arm created / selected?
| This sentence has been rephrased and highlighted green on lines 95 – 98 as:
One hundred and thirteen (113) women who had exited at least one year from the intervention (BioLite and LPG) arms of GRAPHS prior to commencement of this study were purposively selected for the focus group discussions (FGD). Participants included 59 LPG and 54 BioLite users.
|
7. | It would be useful to know how similar the participants/households selected for this study are to those in the GRAPHS study.
| The participants who participated in this study were the same participants who had participated in the GRAPHS but had exited the GRAPHS for one year prior to the start of this current study. |
8. | In Table 1 – there are some differences between the LPG and BioLite groups. This should be discussed in terms of impact on the results reported here.
| We note that the LPG arm participants had a higher percentage of participants who have completed basic education. We believe that this was an artifact of the small sample size of this qualitative study. Since our goal in this study is not to compare across study arms, we are not concerned that this difference introduced bias. Refer to lines 152 – 155 of manuscript. |
9. | The Conclusions section still seems strong given the limitations discussed (and those that I have commented on) – particularly connecting the enablers / barriers to actual use.
| Noted, thank you.. |

Reviewer 2 Report
The authors have not well addressed the result presentation issue. For a journal paper, a more scientific way of summarizing and presenting the results is needed, instead of just directly quoting the responses from interviewees. However, if such a presentation approach is common in this particular area, the paper is worthy to be published.
Author Response
Reviewer 2 | Comment | Response |
The authors have not well addressed the result presentation issue. For a journal paper, a more scientific way of summarizing and presenting the results is needed, instead of just directly quoting the responses from interviewees. However, if such a presentation approach is common in this particular area, the paper is worthy to be published. | Yes, this presentation approach is common in this particular area. Moreover, being a qualitative study, it is necessary that the experiences and views expressed by the participants/respondents are captured in their own words just as reported/narrated rather than being summarized. [1-3]
|
1. Wu, S., D.C. Wyant, and M.W. Fraser, Author guidelines for manuscripts reporting on qualitative research. Journal of the Society for Social Work and Research, 2016. 7(2): p. 405-425.
2. Booth, A., et al., COREQ (consolidated criteria for reporting qualitative studies). Guidelines for reporting health research: a user's manual, 2014: p. 214-226.
3. O’brien, B.C., et al., Standards for reporting qualitative research: a synthesis of recommendations. Academic Medicine, 2014. 89(9): p. 1245-1251.
